# Unique Glycoform-Dependent Monoclonal Antibodies for Mouse Mucin 21

**DOI:** 10.3390/ijms23126718

**Published:** 2022-06-16

**Authors:** Jun Nishida, Shigeyuki Shichino, Tatsuya Tsukui, Mayumi Hoshino, Tomoko Okada, Kyoko Okada, Yuri Yi, Seiko Toraya-Brown, Miho Mochizuki, Ryouta Koizumi, Katrin Ishii-Schrade, Kaori Denda-Nagai, Tatsuro Irimura

**Affiliations:** 1Laboratory of Cancer Biology and Molecular Immunology, Graduate School of Pharmaceutical Sciences, The University of Tokyo, 7-3-1 Hongo, Bunkyo-ku, Tokyo 113-0033, Japan; nishijun0826@gmail.com (J.N.); s_shichino@rs.tus.ac.jp (S.S.); tatsuya.tsukui@ucsf.edu (T.T.); cachecacheflower@gmail.com (M.H.); itomokoo25@gmail.com (T.O.); kyontohapi@yahoo.co.jp (K.O.); yuriyi@hotmail.com (Y.Y.); seiko.toraya@gmail.com (S.T.-B.); miho.mochizuki@riken.jp (M.M.); koizumi.ryouta2@kao.co.jp (R.K.); ishii-schrade@juntendo.ac.jp (K.I.-S.); t-irimura@juntendo.ac.jp (T.I.); 2Division of Molecular Regulation of Inflammatory and Immune Diseases, Research Institute of Biomedical Sciences, Tokyo University of Science, 2641 Yamazaki, Noda-shi, Chiba 278-0022, Japan; 3Division of Glycobiologics, Department of Breast Oncology, Faculty of Medicine, Juntendo University, 2-1-1 Hongo, Bunkyo-ku, Tokyo 113-8421, Japan; 4Intractable Disease Research Center, Graduate School of Medicine, Juntendo University, 2-1-1 Hongo, Bunkyo-ku, Tokyo 113-8421, Japan

**Keywords:** glycoform, monoclonal antibody, mucin, Muc21

## Abstract

Mucin 21(Muc21)/epiglycanin is expressed on apical surfaces of squamous epithelia and has potentially protective roles, which are thought to be associated with its unique glycoforms, whereas its aberrant glycosylation is implicated in the malignant behaviors of some carcinomas. Despite the importance of glycoforms, we lack tools to detect specific glycoforms of mouse Muc21. In this study, we generated two monoclonal antibodies (mAbs) that recognize different glycoforms of Muc21. We used membrane lysates of Muc21-expressing TA3-Ha cells or Chinese hamster ovary (CHO)-K1 cells transfected with *Muc21* as antigens. Specificity testing, utilizing Muc21 glycosylation variant cells, showed that mAb 1A4-1 recognized Muc21 carrying glycans terminated with galactose residues, whereas mAb 18A11 recognized Muc21 carrying sialylated glycans. mAb 1A4-1 stained a majority of mouse mammary carcinoma TA3-Ha cells in vitro and in engrafted tumors in mice, whereas mAb 18A11 recognized only a subpopulation of these. mAb 1A4-1 was useful in immunohistochemically detecting Muc21 in normal squamous epithelia. In conclusion, these mAbs recognize distinct Muc21 epitopes formed by combinations of peptide portions and *O*-glycans.

## 1. Introduction

Mucins are highly glycosylated proteins with a large molecular weight and which often contain tandem repeats that are rich in threonine and serine. They are expressed on epithelial cell surfaces and are believed to play an important role in epithelial defense. They are also involved in several other functions, such as epithelial cell renewal, cell differentiation, cell adhesion, and the transmission of substances to mucus epithelial cells [1,2]. To date, 21 human mucin genes have been identified. The expression of some of them appears to be organ-specific, but overlapping and redundant expression profiles are also common features. Some mucins are associated with cancer pathogenesis by masking tumor antigens and avoiding immune surveillance, modifying cell-to-cell communication, inducing inflammatory microenvironments, and conferring resistance to therapeutic intervention [3].

Recently, we identified a novel transmembrane mucin, Muc21, and implicated it as the molecular entity of epiglycanin, a mucin that has been studied extensively for nearly 50 years [4]. Epiglycanin was first reported in 1975 as a mucin-type cell surface glycoprotein [5]. It was expressed by TA3-Ha cells, a subline of murine mammary carcinoma TA3 cells [6]. An astonishing property reported for TA3-Ha cells was that they could grow in and kill allogeneic mouse strains, whereas TA3-St cells, which did not express epiglycanin, could only grow in syngeneic hosts [6]. Since we identified the epiglycanin/Muc21 gene in mice and found its orthologue in humans, we have made attempts to elucidate the functions of Muc21. For example, the tandem repeat portion of Muc21 plays a crucial role in cell adhesion by interfering with the function of integrin in mice [7]. In addition, glycoforms of human MUC21 can be used as a marker to distinguish differentiated and undifferentiated esophageal epithelial cells [8]. Furthermore, malignant esophageal epithelial cells only express poorly glycosylated MUC21 [8]. Several mAbs against MUC21 bind to human MUC21 glycoforms by recognizing epitopes formed by combinations of peptides and *O*-glycans [8]. Recent studies have shown that MUC21 acts a negative marker for epithelioid mesothelioma [9] and that MUC21 proteins with a specific glycosylation status may be involved in the progression of EGFR-mutated lung adenocarcinomas [10]. Finally, we reported that the expression of MUC21 confers resistance to apoptosis in an *O*-glycosylation-dependent manner, underscoring the glycoform-dependent functions of mucins [11]. Although monoclonal antibodies specific to several glycoforms of human MUC21 have been previously established, antibodies for mouse Muc21 have not been previously obtained.

Here, we aimed at establishing monoclonal antibodies (mAb) recognizing distinct epitopes formed by combinations of mouse Muc21 polypeptide and *O*-glycans. We injected cell membrane fractions of TA3-Ha cells or intact Muc21-expressing CHO-K1 cells into hamsters and used a panel of Muc21-transfected cells, each expressing a different Muc21 glycoform, to screen hybridomas. Using this strategy, we were able to obtain one mAb that is specific to Muc21 with asialo-*O*-glycans, Thomsen–Friedenreich (T) antigen, (T-Muc21), and one mAb that is specific to Muc21 with sialylated *O*-glycans (sialylated T-Muc21). These antibodies were found to be useful for the elucidation of the heterogeneous expression of Muc21 in in vivo-generated tumors and in normal mouse tissues.

## 2. Results

### 2.1. Preparation of Anti-Mouse Muc21 mAbs

We aimed to prepare mAbs specific to mouse Muc21/epiglycanin with different O-linked glycans, in particular, to prepare an mAb which recognizes Muc21 with truncated glycans and another mAb which recognizes Muc21 with extended glycans. Such antibodies could be useful in differentiating Muc21 expressed in various organs and in cells at different stages of differentiation of these organs, as shown with anti-human MUC21 antibodies [8]. Anti-mouse Muc21 antibodies could be useful in eventually exploring the unique glycoform-associated functions of Muc21.

To obtain antibodies specific to Muc21 with truncated glycans, repeated immunizations of hamsters were performed with lysates of membrane fractions from TA3-Ha cells. After an elevated antibody titer for benzyl-GalNAc-treated CHO-K1 cells transfected with N-FLAG-Muc21 cDNA by ELISA was confirmed in hamster sera, splenocytes were obtained and fused with mouse myeloma cells, PAI. Hybridomas were prepared and first screened via flow cytometry with CHO-K1-pcDNA3.1-Muc21 cells treated with benzyl-GalNAc. The second screening was performed by means of flow cytometry with TA3-Ha and TA3-St cells, as well as with CHO-K1-Mock and CHO-K1-Muc21 cells both with and without benzyl-GalNAc treatment, which resulted in two hybridoma wells specific to TA3-Ha cells and CHO-K1-pcDNA3.1-Muc21 cells treated with benzyl-GalNAc. The third screening process was carried out via immunohistochemistry of normal mouse tissues. After hybridoma cloning, mAb 1A4-1 was obtained. The immunization and screening process for mAb 1A4-1 is summarized in the left panel of Figure 1.

For the preparation of antibodies specific to Muc21 with extended glycans, repeated immunizations of hamsters with CHO-K1-pCAGGS-*Muc21* cells were performed. Hybridomas were first screened via ELISA with purified N-FLAG-Muc21, and a second screening was performed via flow cytometric analyses with CHO-K1-pcDNA3.1-N-FLAG-*Muc21* cells. The third screening was performed via Western blotting with B16-pcDNA-*Muc21* cells and flow cytometric analysis with CHO-K1-pcDNA3.1-N-FLAG-*Muc21* and CHO-Lec2-FLAG-*Muc21* cells. One of 11 hybridoma wells resulted in mAb 18A11. The immunization and screening process for mAb 18A11 is summarized in the right panel of Figure 1.

### 2.2. Examination of Binding Specificity of mAb 1A4-1 and mAb 18A11 via Flow Cytometric Analysis

To characterize the binding specificity of mAbs 1A4-1, we performed flow cytometric analysis and Western blotting analysis with three CHO glycoform variants which express Muc21, namely, CHO-K1-pcDNA3.1-N-FLAG-*Muc21* cells, CHO-Lec2-pCAGGS-N-FLAG-*Muc21* cells, and CHO-ldlD-pCAGGS-N-FLAG-*Muc21* cells. Sialidase-treated CHO-K1-pcDNA3.1-N-FLAG-*Muc21* cells were also used. The putative terminal carbohydrate structures on Muc21 expressed by the CHO variants used are depicted in Figure 2. Actual terminal glycosylation structures present on these CHO variant cells were confirmed via Western blotting with anti-FLAG-M2 mAb and via lectin blotting with *Vicia villosa* (VVA), *Arachis hypogaea* (PNA), and *Wheat germ agglutinin* (WGA) lectins.

In our flow cytometric analysis, using CHO-K1-pcDNA3.1-N-FLAG-*Muc21* cells and their mock transfectants, hamster IgG and mAb 1A4-1 did not bind to mock transfectants or Muc21 transfectants. Anti-FLAG mAb and mAb 18A11 bound to CHO-K1-pcDNA3.1-N-FLAG-*Muc21* cells but not to mock cells (Figure 3a). CHO-Lec2-pCAGGS-N-FLAG-*Muc21* cells provided interesting information. mAb 1A4-1 strongly bound to CHO-Lec2-pCAGGS-N-FLAG-*Muc21* cells, whereas mAb 18A11 did not (Figure 3b). These results strongly suggest that mAb 18A11 binds to sialylated T-Muc21 and mAb 1A4-1 binds to T-Muc21. These assumptions were confirmed through a comparison of antibody bindings before and after sialidase treatment of CHO-K1-pcDNA3.1-N-FLAG-*Muc21* cells. mAb 18A11 did not bind CHO-Lec2-pCAGGS-N-FLAG-*Muc21* cells or sialidase-treated CHO-K1-pcDNA3.1-N-FLAG-*Muc21* cells. These cells express glycans without sialic acids, so T-antigens are present (Figure 3c). In the analysis using CHO-ldlD-pCAGGS-N-FLAG-*Muc21* cells, hamster IgG, mAb 1A4-1 and mAb 18A11 bound to Muc21 transfectant to the same extent as to the mock transfectant (Figure 3d). This result suggests that both mAb 1A4-1 and mAb 18A11 do not bind to unmodified Muc21.

To investigate the inter-species cross reactivity of these mAbs, we also analyzed the binding of mAbs to human *MUC21*-transfected CHO cell variants via flow cytometry. The binding of mAb 1A4-1 to CHO-Lec2 transfected with human *MUC21* was mostly equal to that of mock cells, whereas mAb heM21C (which is specific to Tn, T, and sialyl T-MUC21 [8]) bound strongly (Figure 3e). mAb 18A11 bound very weakly to CHO-K1 transfected with human *MUC21* (Figure 3f). Therefore, to test if mAb 18A11 merely recognized sialyl T-carbohydrate chains, B16-F10 cells transfected with human *MUC1* were used. No binding of mAb 18A11 was observed, whereas mAb MY1E12, which is specific to sialyl-T-MUC1 [13], bound strongly (Figure 3g).

### 2.3. Examination of Binding Specificity of mAb 1A4-1 and mAb 18A11 via Western and Lectin Blotting Analysis

Western blotting and lectin blotting analysis also supported the above conclusions. Cell lysates were separated by means of sodium dodecyl sulfate-polyacrylamide gel electrophoresis (SDS-PAGE) on 4% gels. After SDS-PAGE and blotting, anti-FLAG mAb, mAb 18A11, and mAb 1A4-1 were reacted on the membranes. Lectins were used alongside mAbs to confirm the presence or absence of glycan structures. In this analysis, we prepared lysates from CHO-K1 cells, Lec2 cells, and ldlD cells transfected with N-FLAG-*Muc21*. CHO-K1 cells were treated either with sialidase alone or with both sialidase and β-galactosidase. As determined by lectin blotting analysis, the presence of sialic acid on Muc21 of CHO-pcDNA3.1-N-FLAG-*Muc21* cells was confirmed by the fact that *Wheat germ agglutinin* (WGA), recognizing sialic acid and *Maackia amurensis* hemagglutinin (MAH), recognizing disialyl T, bound to the immunoprecipitated cell lysate (Figure 4a and data not shown). Similarly, the binding of PNA, having affinity to Galß1-3GalNAc, to immunoprecipitated cell lysates showed that sialidase-treated CHO-K1-pcDNA3.1-N-FLAG-*Muc21* cells and CHO-Lec2-pCAGGS-N-FLAG-*Muc21* cells expressed T-Muc21 (Figure 4a,b). VVA, with affinity for GalNAc, was used to confirm that a part of CHO-Lec2-pCAGGS-N-FLAG-*Muc21* cell lysates and CHO-K1-pcDNA3.1-N-FLAG-*Muc21* cell lysates treated with sialidase and β-galactosidase expressed Tn-Muc21 (Figure 4a,b). Muc21 with terminal sialic acid was detected via Western blotting at a migration distance corresponding to an apparent molecular weight of around 200 kDa, whereas other less negatively charged Muc21 glycoforms were detected above 200 kDa. In Western blotting analysis with CHO-K1-pcDNA3.1-N-FLAG-*Muc21* cell lysates (Figure 4a), mAb 18A11 bound to Muc21 without any enzymatic treatment, whereas it did not bind to Muc21 treated with sialidase or both sialidase and β-galactosidase. In contrast, mAb 1A4-1 bound to sialidase-treated Muc21 lysates, and this binding weakened after combined sialidase and β-galactosidase treatment. mAb 1A4-1 also bound to CHO-Lec2-pCAGGS-N-FLAG-*Muc21* cells, whereas mAb 18A11 did not (Figure 4b). Both mAb 1A4-1 and mAb 18A11 did not recognize lysates of CHO-ldlD-pCAGGS-N-FLAG-*Muc21* cells (Figure 4c). In conclusion, mAb 1A4-1 and mAb 18A11 specifically recognize distinct glycoforms of Muc21. It was shown that mAb 1A4-1 is specific to T-Muc21, whereas mAb 18A11 is specific to sialylated T-Muc21.

### 2.4. Immunohistological Staining of Breast Tumors Formed via the Injection of TA3 Mammary Carcinoma Cells

The ability of mAbs 1A4-1 and 18A11 to bind Muc21, expressed by TA3-Ha and TA3-St cells in culture and in situ, was tested. As determined via flow cytometric analysis, binding of these antibodies was not observed with TA3-St cells, whereas binding of these antibodies was observed with TA3-Ha cells. The binding profile of mAb 1A4-1 to TA3-Ha cells was uniform, whereas mAb 18A11 stained only a portion of the TA3-Ha cells (Figure 5a). Breast tumors were induced in A/J mice via the injection of TA3 mammary carcinoma cells into mammary fat pads. The formation of tumors was observed after injections into mammary fat pads of both TA3-Ha and TA3-St cells. H&E staining (Figure 5b,f) and immunohistochemical staining of tumor tissue were performed (Figure 5c–e,g–i). mAb 1A4-1 and mAb 18A11 did not bind to breast tumors formed by TA3-St cells (Figure 5d,e). Tumors derived from TA3-Ha cells were stained with both antibodies, and the binding of mAb 18A11 appeared to be heterogeneous (Figure 5h,i). These results indicated that breast cancer tissue derived from TA3-Ha cells expresses both Muc21 with T-antigen and sialylated T-antigen.

### 2.5. Muc21 mRNA Expression in Normal Mouse Tissues and Immunohistochemical Staining of Normal Mouse Tissues with mAbs 1A4-1 and 18A11

To investigate the mRNA expression of *Muc21*, we performed RT-PCR using a panel of normal mouse tissues. As a result, *Muc21* mRNA expression was observed in tissues characterized by squamous epithelia, including vagina, esophagus, eye, stomach, and thymus tissues (Figure 6a). Based on these results, we decided to investigate whether mAbs 1A4-1 and 18A11 can detect Muc21 protein in the esophagus and vagina via immunohistochemistry using large intestine tissue as a negative control.

With mAb 18A11, no antibody binding could be detected in any of the mouse tissues tested (data not shown). With mAb 1A4-1, antibody binding could only be observed after the tissue had been treated with sialidase. In sialidase-treated esophagus and vagina tissue, antibody binding was observed on the luminal side of the squamous epithelium (Figure 6b,c). No antibody binding after sialidase treatment was observed in the large intestine tissue, which served as a negative control (Figure 6d).

## 3. Discussion

We prepared two new mAbs, 1A4-1 and 18A11, that specifically recognize Muc21 with distinct glycoforms, and used them to profile the glycan structures of Muc21 on TA3-Ha cells in vitro in culture and in situ in tumor tissue. We found that mAb 18A11 bound to sialylated T-Muc21, but not to Muc21 carrying T or Tn-antigen, and not to non-*O*-glycosylated Muc21, whereas mAb 1A4-1 bound to T-Muc21. Immunostaining showed that both glycoforms, sialylated T-Muc21 and T-Muc21, were present in TA3-Ha cells in vitro and in situ, suggesting no major change in the expression profiles of the two glycoforms during tumor formation. Muc21 with both sialylated and non-sialylated T-antigen was expressed on TA3-Ha cells, although some cells were negative for sialylated T-Muc21 both in vitro and in situ. These results imply that the glycan structures on Muc21 expressed by TA3-Ha cells exhibit some degree of heterogeneity. Considering the possibility that some TA3-Ha cells have both T-Muc21 and sialylated T-Muc21, our results are in line with previous findings by Codington and co-workers showing that the carbohydrate composition of epiglycanin isolated from murine ascites fluid was composed of 60% T-antigen and 13% sialyl T-antigen [14].

The strength of the present study lies in the use of a multi-step screening process, including CHO-variant cells which express different glycoforms of Muc21. This screening process enabled us to obtain mAbs with distinct protein-carbohydrate epitope specificities, T-Muc21 and sialylated T-Muc21, respectively. The use of TA3-Ha and CHO-K1-*Muc21* cells should theoretically cover a wide range of Muc21 glycoforms, including Tn-Muc21 and non-glycosylated Muc21, as immunogens. However, the abundance and immunogenicity of different Muc21 glycoforms in these immunogens is likely variable. In the future, obtaining monoclonal antibodies against all of the possible Muc21 glycoforms may possibly maximize the spectrum of research applications.

We immunohistochemically stained normal mouse tissues with mAbs 1A4-1 and 18A11. We found that mAb 18A11 did not show any staining in any of the tissues tested, although the same antibody was able to stain TA3-Ha cells in situ. In contrast, mAb 1A4-1 was useful for the staining of normal mouse esophagus and vagina tissues after the tissue had been treated with sialidase. In these tissues, the binding could be observed on the luminal side of the squamous epithelium. These results suggest that sialylated T-Muc21 is expressed in normal mouse esophagus and vagina tissues. Since the epitope structure or its accessibility is somewhat different from that on TA3-Ha cells, mAb 18A11 is not able to bind Muc21 expressed in normal mouse tissues. Consistent with the mRNA expression pattern, antibody binding of mAb 1A4-1 was seen in esophagus and vagina tissues, with no staining in tissues of the large intestine. These results are also in agreement with the organ distribution of human MUC21 protein, as reported in the *Human Protein Atlas* and as previously published [8].

Previous studies on epiglycanin in mouse experimental systems suggested that the expression of this molecule is associated with highly malignant phenotypes [15,16]. Evidence supporting the hypothesis that glycoforms are important in determining malignant behavior was recently provided in pathological investigations using glycoform-specific mAbs for human MUC21 [17]. Similar associations with other membrane-type mucins, such as MUC1 and MUC4, have been widely claimed. For example, aberrantly glycosylated MUC1 is overexpressed in breast cancer and in other cancers [18,19]. Cancer-associated *O*-glycans are often shortened and commonly contain Tn and T-antigens and their sialylated versions [20]. The presence of Tn and T-antigens has been implicated in relation to carcinoma aggressiveness [20,21]. Furthermore, sialylated T-antigen was associated with an enhanced growth rate of mammary carcinoma cells in MUC1 transgenic mice [22] and with the malignancy of bladder cancer [23], ovarian cancer [24], and colorectal cancer [25] in humans. Therefore, sialylated and non-sialylated T, as well as Tn-antigen, are associated with enhanced malignant behaviors. On the other hand, we showed that sialic acid on MUC1 potentially exerts inhibitory effects on the peritoneal dissemination of clear cell-type ovarian cancer cells in a mouse model [26]. Thus, there is accumulating evidence that the glycoforms of mucins determine their functions, and tools that specifically and efficiently identify distinct glycoforms of mucins and other tumor cell antigens will advance cancer treatments.

With regard to the importance of sialylation in cancer malignancy, recent reports indicated that sialylation regulated the stability of stem cells. For example, α2,3-sialylation contributed to the stability of the CD133 protein in defining tumor-initiating cells [27]. Other reports showed that MUC1 present on side-population cells found in the MCF7 breast cancer cell line was heavily sialylated and carried sialyl T-antigen [28] and that *O*-linked alpha2,3 sialylation defines stem cell populations in breast cancer [29]. In addition, non-sialylated T-antigen is immunogenic, whereas sialylated mucins are poorly immunogenic [21], resulting in their escape from immune surveillance. Furthermore, by knocking down β-galactoside α2-6-sialyltransferase ST6Gal-1, which catalyzes the addition of sialic acid to galactose, ovarian tumor cells gained cisplatin resistance [30]. Together with a recent study showing that the overexpression of human T-MUC21 and sialyl T-MUC21 confers anti-apoptotic properties to cells, whereas non-glycosylated MUC21 and Tn-MUC21 do not [11], these findings underscore the importance of glycoforms in the regulation of mucin function. The 1A4-1 and 18A11 mAbs will potentially be useful tools to elucidate this structure–function relationship in Muc21. This will expand our understanding of how mucins and their differential glycosylation are involved in epithelial protection and cancer malignancy. Furthermore, these antibodies and similar antibodies for glycoforms of human MUC21 may serve as seed compounds useful in the development of therapeutic antibodies.

## 4. Materials and Methods

Animals: Female 5-week-old A/J mice were bought from Nihon SLC Inc. (Shizuoka, Japan) and housed under specific pathogen-free conditions. Six-week-old male/female Armenian hamsters were bought from Oriental Yeast Co., Ltd. (Shizuoka, Japan) and housed under conventional conditions according to the guidelines of the Animal Use and Care Committee of the Graduate School of Pharmaceutical Sciences of The University of Tokyo.

Cells and cell cultures: CHO-K1 cells and CHO-Lec2 [31] cells were obtained from the American Type Culture Collections. TA3-Ha cells and TA3-St cells were kind gifts from Dr. J. F. Codington. All cells were cultured in a 1:1 mixture of Dulbecco’s modified minimum essential medium (Nissui, Tokyo, Japan) and Ham’s F12 (Nissui) with 10% fetal calf serum (D/F 10% FCS) at 37 °C in a 5% CO_2_ atmosphere. CHO-ldlD cells [32], a kind gift from Dr. M. Krieger (Massachusetts Institute of Technology, Cambridge, MA, USA), were cultured in D/F with 10% dialyzed FCS at 37 °C in a 5% CO_2_ atmosphere. PAI myeloma cells, kindly provided by Dr. J. Aoki (Graduate School of Pharmaceutical Sciences, The University of Tokyo, Tokyo, Japan), and newly established hybridoma cells were cultured in GIT medium (Wako, Tokyo, Japan) with 5% NCTC109 medium (Gibco, Grand Island, NY, USA), 1% MEM NEAA (Gibco) and 2 mM L-glutamine (Mediatech, Manassas, VA, USA). CHO-pcDNA3.1-N-FLAG-*Muc21* cells used for mAb 1A4-1 screening were cultured in D/F 10% FCS with 2 mM benzyl-*N*-acetylgalactosaminide (benzyl-GalNAc) (Sigma-Aldrich; Merck, Darmstadt, Germany). Benzyl-GalNAc treatment was used to truncate *O*-glycans [33]. B16-F10-*MUC1* cells were prepared and cultured as previously reported [34].

Muc21-transfectants: For the preparation of CHO-K1-*Muc21* transfectants (used for immunization of hamsters) and for the generation of CHO-Lec2-N-FLAG-*Muc21* and CHO-ldlD-N-FLAG-*Muc21* transfectants (used for hybridoma screening and antibody specificity investigation), the coding sequence of *Muc21* containing 84 tandem repeats (84 TR) was inserted into the pCAGGS-neo vector. Transfection was performed with Lipofectamine LTX with Plus Reagent (Invitrogen; Thermo Fisher Scientific, Waltham, MA, USA) following the protocol of the manufacturer. Transfectants were selected with up to 400 μg/mL Geneticin (G418 sulfate, Calbiochem; Merck, Darmstadt, Germany) and cloned by means of the limiting dilution method. For the preparation of CHO-K1-pcDNA3.1-N-FLAG-*Muc21* transfectants and B16-F1 *Muc21* transfectants (used for hybridoma screening and antibody specificity investigation), the coding sequence of Muc21 containing 84 TR was inserted into the pcDNA3.1 vector (Invitrogen) and transfection and selection were performed as previously described [7]. The putative glycan structures on Muc21 expressed by the transfected cells used in the present study are shown in Figure 2.

TA3-Ha cell membrane lysates: The membrane fraction of TA3-Ha cells was isolated by homogenizing 1 × 10^8^ TA3-Ha cells in 1 mL homogenization buffer (5 mM Tris-HCl, 1 mM EDTA, 1% protease inhibitor cocktail (Sigma-Aldrich; Merck, Darmstadt, Germany), pH 9.0) using 80–100 strokes of an electric homogenizer. The homogenate was centrifuged at 3000 rpm for 10 min at 4 °C. The pellet was discarded and the resulting supernatant again centrifuged at 34,000 rpm for 1 h at 4 °C. The pellet (containing the cell membrane) was re-suspended and dissolved in lysis buffer (10 mM Tris-HCl, 2 mM EDTA, 50 μM CaCl_2_, 0.5% NP-40, 0.1% protease inhibitor cocktail (Sigma-Aldrich), pH 7.2), centrifuged at 15,000 rpm for 5 min at 4 °C and the supernatant was designated as “TA3-Ha membrane lysate”. Total protein was quantified using the BCA assay (Pierce; Thermo Fisher Scientific).

Preparation of mAb 1A4-1: Six-week-old male Armenian hamsters were immunized with TA3-Ha membrane lysate corresponding to 400 μg total protein mixed 1:1 (vol/vol) with Freund’s complete adjuvant (DIFCO; Thermo Fisher Scientific, Waltham, MA, USA) intraperitoneally on day 0, and again using the same immunogen, but mixed 1:1 (vol/vol) with Freund’s incomplete adjuvant on days 14, 29, 37, and 99. Hamsters were sacrificed on day 102 for the isolation of spleen cells. Splenocytes and mouse myeloma cells, PAI, were combined at a 4:1 ratio and fused with 50% polyethylene glycol solution (Sigma-Aldrich). For hybridoma selection, cells were incubated in GIT medium supplemented with HAT mixture (DS Pharma Biomedical, Suita, Japan) and 10% BM-Condimed H1 (Roche, Basel, Switzerland) for 1 week at 37 °C in a 5% CO_2_ atmosphere. Hybridomas were prepared and first screened by flow cytometry with CHO-K1-pcDNA3.1-*Muc21* cells treated with benzyl-GalNAc. The second screening was performed via flow cytometry with TA3-Ha and TA3-St cells, as well as with CHO-K1-Mock and CHO-K1-*Muc21* cells both with and without benzyl-GalNAc treatment. The third screening was carried out via immunohistochemistry of normal mouse tissues. After hybridoma cloning via the limiting dilution method, mAb 1A4-1 was obtained.

Preparation of mAb 18A11: Female conventional Armenian hamsters were immunized intraperitoneally with an emulsion made from 1 × 10^7^ CHO-K1-pCAGGs-neo-*Muc21* cells suspended in 100 μL sterile saline and 100 μL Sigma-Aldrich adjuvant system 5 times at 2-week intervals. Blood was collected before the first immunization and one week after each immunization from the retro-orbital venous plexus. Blood was incubated for 30 min at 37 °C, then at 4 °C overnight, and then centrifuged at 4 °C and at 15,000 rpm for 15 min. Sera were collected and stored at −20 °C for antibody titer checking. After spleen extraction, splenocytes and mouse myeloma cells, PAI, were combined at a 4:1 ratio and fused with 50% polyethylene glycol solution (Sigma-Aldrich). For hybridoma selection, cells were incubated in GIT medium supplemented with HAT mixture (DS Pharma) and 10% BM-Condimed H1 (Roche, Basel, Switzerland) for 1 week at 37 °C in a 5% CO_2_ atmosphere. Hybridomas were first screened by ELISA with purified N-FLAG-Muc21, and a second screening was carried out via flow cytometric analyses with CHO-K1-pcDNA3.1-N-FLAG-*Muc21* cells. The third screening was performed via flow cytometric analysis with CHO-K1-pcDNA3.1-N-FLAG-*Muc21* cells and Western blotting with B16-F1-*Muc21* cells. Hybridoma cloning was performed through the limiting dilution method and mAb 18A11 was obtained.

Purification of N-FLAG-Muc21: CHO-K1-pcDNA3.1-N-FLAG-*Muc21* cell lysate was diluted with TBS to a protein concentration of 0.4 mg/mL. One milliliter of a 50% Sepharose-4B solution was added to 9 mL of cell lysate and rotated for 15 min at 4 °C. The suspension was centrifuged at 4 °C and 1500 rpm for 4 min and the supernatant was collected. The supernatant was passed through a 0.45 µm filter. One milliliter of a 50% anti-FLAG agarose (Sigma) suspension was added to the supernatant and rotated overnight at 4 °C. After centrifugation at 4 °C and 1500 rpm for 4 min, the supernatant was removed and the agarose was washed with TBST. Five hundred microliters of a 100 µg/mL FLAG peptide (Sigma) in TBS solution were added to the agarose and rotated for 10 min at 4 °C. The suspension was centrifuged at 4 °C and 1500 rpm for 4 min and the supernatant was collected. The supernatant was dialyzed against PBS for one day at 4 °C.

ELISA with purified N-FLAG-Muc21 (first screening): Purified N-FLAG-Muc21 in PBS was coated using 100 μL of 1 μg/mL per well on a 96-well flat-bottom plate and incubated overnight at 4 °C. Plates were blocked with 1/10 diluted ImmunoBlock (DS Pharma) in Milli-Q water for 45 min at room temperature. After 3 washes with PBS containing 0.1% Tween-20 (PBST), 100 μL of diluted anti-FLAG-M2 mAb (Sigma-Aldrich), diluted hamster IgG (BioLegend, San Diego, CA, USA), or hybridoma supernatant was added and incubated for 1.5 h at room temperature. Plates were washed 3 times with PBST, and 100 μL of diluted HRP goat anti-hamster IgG (Santa Cruz Biotechnology, Dallas, TX, USA) or HRP goat anti-mouse IgG (Millipore; Merck) was added and incubated for 45 min at room temperature. After plates were washed 5 times with PBST, 100 μL of TMB solution composed of 10% TMB-DMSO, 1/1000 30% H_2_O_2_ in 0.05 M phosphate citrate buffer (pH 5.0) was added and incubated for 10 min at room temperature. Absorbance was measured at 450 nm/630 nm with an ARVO X5 plate reader (PerkinElmer Japan, Kanagawa, Japan).

ELISA with CHO-pcDNA3.1-N-FLAG-Muc21 cells (second screening): Cells were plated at 5 × 10^4^ cells per well onto 96-well flat-bottom plates and incubated for 1 day at 37 °C in a 5% CO_2_ atmosphere. After washing 3 times with PBS, cells were dried for 10 min at room temperature and fixed with 100% ethanol for 1 min at 4 °C. Then, after ethanol removal and drying for 10 min at room temperature, cells were blocked with 200 μL of 2% bovine serum albumin (BSA)-PBS overnight at 4 °C. After 3 washes with PBST, 100 μL of diluted anti-FLAG-M2 mAb (Sigma-Aldrich), diluted hamster IgG (BioLegend), or hybridoma supernatant was added and incubated for 1.5 h at room temperature. Plates were washed 3 times with PBST, and 100 μL of diluted HRP goat anti-hamster IgG (Santa Cruz Biotechnology) or HRP goat anti-mouse IgG (Millipore) was added and incubated for 45 min at room temperature. After plates were washed 5 times with PBST, 100 μL of TMB solution composed of 10% TMB-DMSO, 1/1000 30% H_2_O_2_ in 0.05 M phosphate citrate buffer (pH 5.0) was added and incubated for 10 min at room temperature. Absorbance was measured at 450 nm/630 nm with an ARVO X5 plate reader (PerkinElmer).

Flow cytometric analysis (third screening): Cells were collected from cell culture plates via incubation with PBS containing 0.02% ethylenediaminetetraacetic acid (EDTA) for 5 min and suspended in FACS buffer composed of 0.1% BSA, 0.1% NaN_3_ in PBS. Cells were blocked with 10% normal goat serum (NGS)/FACS buffer for 10 min at 4 °C. Anti-FLAG-M2 mAb, hamster IgG or hybridoma supernatant were added and reacted for 30 min at 4 °C. After washing with FACS buffer, FITC goat anti-mouse IgG (Invitrogen) or FITC goat anti-hamster IgG (Southern Biotech, Birmingham, AL, USA) was added and reacted for 30 min at 4 °C. After washing with FACS buffer and removing the supernatant, cell pellets were suspended in FACS buffer and analyzed on an Epics Coulter XL flow cytometer (Beckmann Coulter, Irvine, CA, USA) to determine the level of antibody reactivity. Data were analyzed using FlowJo software (BD).

Sialidase treatment of Muc21 transfectants: Cells were suspended in phosphate buffer (pH 5.8) and incubated with 100 mU of sialidase from *Clostridium perfringens* (Sigma-Aldrich) for 30 min at 37 °C in a 5% CO_2_ humidified atmosphere. Cell pellets were washed with PBS.

β-galactosidase treatment of Muc21 lysate: After sialidase treatment and immunoprecipitation with anti-FLAG-M2 agarose (Sigma-Aldrich), beads were washed with potassium phosphate buffer (pH 6.0) twice and then incubated with 500 mU/mL β-galactosidase from *Bacillus circulans* [35] in potassium phosphate buffer (pH 6.0) at 37 °C overnight. After the reaction, beads were washed with TBS twice.

Immunoprecipitation and Western/lectin blotting analysis: Cells were solubilized in lysis buffer (10 mM Tris pH 7.5, 0.25 M sucrose, 0.25 mM CaCl_2_, 2 mM EDTA, 0.5% NP-40) containing 1% protease inhibitor cocktail (Merck, Darmstadt, Germany). For immunoprecipitation, cell lysates were rotated overnight at 4 °C, following administration of anti-FLAG-M2 agarose (Sigma-Aldrich). Cell lysates were added to sample buffer containing 2-mercaptoethanol, boiled for 5 min at 95 °C, and separated via SDS-PAGE on a 4% polyacrylamide gel. After electrophoresis, samples were transferred to polyvinylidene fluoride (PVDF) membrane (Millipore) and blocked with 3% BSA-PBS overnight at 4 °C. Strips of the membrane were individually reacted with anti-FLAG-M2, hybridoma supernatant, biotin-conjugated VVA (0.5 μg/mL; Sigma-Aldrich), or biotin-conjugated PNA (0.5 μg/mL; Seikagaku, Tokyo, Japan) for 2 h at room temperature, or, alternatively, with biotin-conjugated WGA (2.5 μg/mL; Seikagaku, Tokyo, Japan) overnight at 4 °C. After washing with PBST, membranes were reacted with HRP goat anti-mouse IgG (Zymed; Thermo Fisher Scientific), HRP goat anti-hamster IgG (Santa Cruz Biotechnology) or HRP streptavidin for 1 h at room temperature. Blots were visualized using solutions of Luminol chemiluminescence reagent (Santa Cruz Biotechnology) and detected using an ImageQuant Las4010 instrument (GE Healthcare, Chicago, IL, USA).

Transplantation of TA3-Ha and TA3-St cells in A/J mice and immunohistochemical staining of tumor tissue: TA3-Ha cells and TA3-St cells were washed with PBS twice and suspended in HANKS’ Buffer containing HANKS’ solution (Nissui) with NaHCO_3_, pH 7.4. TA3-Ha or TA3-St cells were injected at 1 × 10^5^ cells per mouse into the mammary fat pad. Thirteen days after injection, mice were sacrificed. Frozen tissues from sacrificed mice were fixed in 10% formalin in PBS for 1 day, 10% sucrose in PBS for 1 day, and 20% sucrose in PBS for 1 day at 4 °C. After fixation, tissues were embedded in Tissue-Tek OCT compound (Sakura Finetek, Tokyo, Japan) and 6 μm-thick cryosections were cut on a cryostat (Leica Microsystems, Wetzlar, Germany). Sections were mounted on Poly-L-lysine-coated glass slides. After washing with PBS, non-specific binding was blocked with 1% H_2_O_2_ PBS and 2% NGS/3% BSA PBS with avidin (Vector Laboratories, Burlingame, CA, USA) for 30 min each at room temperature. Sections were incubated with 1A4-1 mAb, 18A11 mAb, or hamster IgG with biotin (Vector Laboratories) overnight at 4 °C. After washing with PBS, sections were incubated with biotin anti-Hamster IgG for 30 min. After washing with PBS, sections were incubated with HRP-streptavidin and stained using an AEC kit (Invitrogen). Slides were washed with water, counter-stained with hematoxylin, and mounted with 20% polyvinyl alcohol.

RNA isolation and Muc21 RT-PCR using normal mouse tissues: Tissues were removed from C57BL/6J mice and instantly frozen in liquid nitrogen. Frozen tissue was ground in a mortar cooled in liquid nitrogen in 1 mL of ULTRASPEC RNA (BIOTECX, Houston, TX, USA), transferred to a tube, and vortexed. After adding 200 μL of chloroform and mixing via inversion for 15 s, the mixture was allowed to stand on ice for 5 min and centrifuged at 15,000 rpm and 4 °C for 15 min. The upper layer was transferred to another tube, an equal amount of isopropanol was added, and the mixture was inverted 6 times and then allowed to stand on ice for 10 min. The sample was centrifuged at 15,000 rpm at 4 °C for 10 min and the supernatant was discarded. The pellet was rinsed with 500 μL of 75% ethanol and centrifuged at 15,000 rpm at 4 °C for 5 min. After removing the supernatant and volatilizing the remaining ethanol in a heat block warmed to 65 °C, 13 μL of RNAse-free water was added, and the mixture was heated at 65 °C for 1 min to dissolve. OD260/280 was measured and the sample was used as a total RNA solution. Reverse transcription was performed using Superscript III reverse transcriptase (Invitrogen). Two μg total RNA, 1 μL 10 mM dNTP, and 1 μL 0.5 mg/mL oligo dT and milliQ were combined to a total volume of 12 μL. After heating at 65 °C for 5 min, 2 μL 0.1 M DTT, 4 μL 5× first strand buffer, 1 μL anti-RNase, and 1 μL Superscript II were added. The sample was incubated at 42 °C for 50 min and then heated at 70 °C for 15 min. PCR was performed using a 5- to 20-fold diluted cDNA solution as a template. The sequences of the primers for Muc21 are shown below.

Muc21 (421 bp,63 °C)3′epi S/AS_forward: TTCCAGCTCTAGCCTGAGTGCCACCC3′epi S/AS_reverse: ATCTTTGTCAGGGATGTCAGGGACCCG

The composition of the template cDNA solution was 5 μL, 2×Ampli Taq Master Mix 10 μL, 10 μM 3′epi S/AS_forward 0.1 μL, 10 μM 3′epi S/AS_reverse 0.1 μL, DMSO 1 μL, and milliQ 3.8 μL. The reaction conditions were 95 °C for 11 min, (94 °C for 30 s, 63 °C for 45 s, 72 °C for 45 s) × 45, and 72 °C for 7 min.

Sialidase treatment and immunohistochemical staining of mouse tissues: Six-micrometer-thick cryosections of OCT compound-embedded tissues were air-dried, fixed with ethanol for 30 s and then washed with PBS twice for 5 min each. Tissue sections were incubated with sialidase from *Clostridium perfringens* (1 U/mL, Sigma-Aldrich) in 0.1 M sodium acetate buffer containing 10 mM CaCl_2_, pH 5.0 at 37 °C overnight. Sections were washed three times for 5 min with PBS and then blocked with Avidin-D (Vector Laboratories) diluted in 2% NGS/3% BSA PBS for 30 min at room temperature. After washing with PBS three times, the sections were blocked with Biotin (Vector, Laboratories) diluted in 2% NGS/3% BSA PBS for 30 min at room temperature. After washing with PBS three times, sections were incubated with mAb 1A4-1 (1:200), hamster IgG (10µg/mL, MP Biochemicals, Irvine, CA, USA) diluted in 2% NGS/3% BSA PBS, or 2% NGS/3% BSA PBS for 1 h at room temperature. Sections were washed three times with PBS and then incubated with biotin-conjugated goat anti-Hamster IgG (1:100) diluted in 2% NGS/3% BSA PBS for 30 min at room temperature. After washing five times for 10 min each with PBS, sections were incubated with 0.3% H2O2 PBS for 30 min at room temperature, and then washed three times with PBS. Sections were incubated with HRP-conjugated streptavidin (Zymed) (1:100) diluted in PBS for 30 min at room temperature. After washing three times with PBS, color was developed with an AEC kit (Invitrogen). Serial tissue sections not treated with sialidase were stained in the same way as sialidase-treated sections. Slides were washed with water and mounted with 20% polyvinyl alcohol.

## 5. Conclusions

We generated two mAbs, 18A11 and 1A4-1, specifically recognizing sialylated T-Muc21 and T-Muc21, respectively. These two antibodies represent the first tools for distinguishing glycoforms of Muc21, and they might be useful for understanding the role of Muc21 in health and disease.

## Figures and Tables

**Figure 1 ijms-23-06718-f001:**
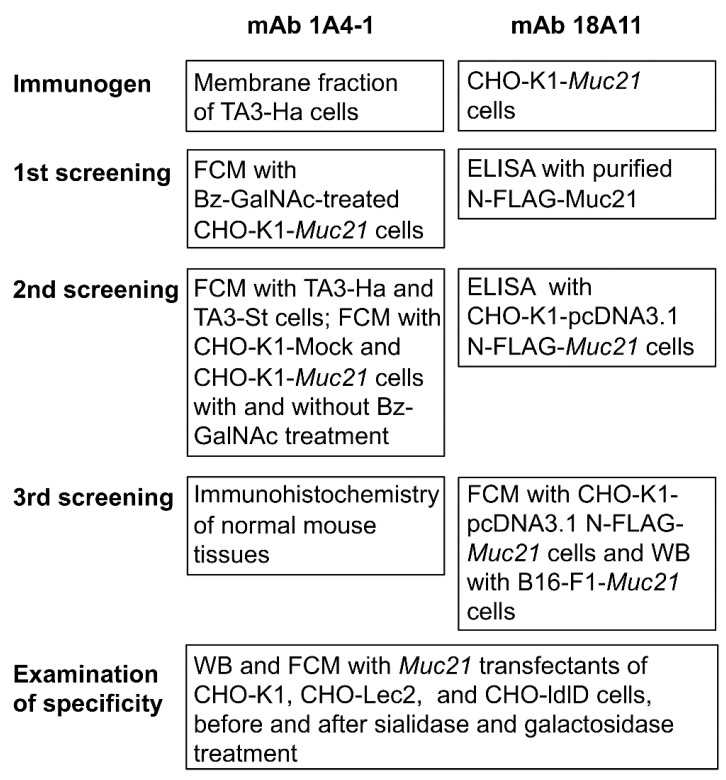
Strategy to obtain monoclonal antibodies (mAbs) 1A4-1 and 18A11, which are specific to distinct glycoforms of Muc21. Bz-GalNAc; benzyl-*N*-acetylgalactosaminide, CHO; Chinese hamster ovary cells, FCM; flow cytometry, WB; Western blotting.

**Figure 2 ijms-23-06718-f002:**
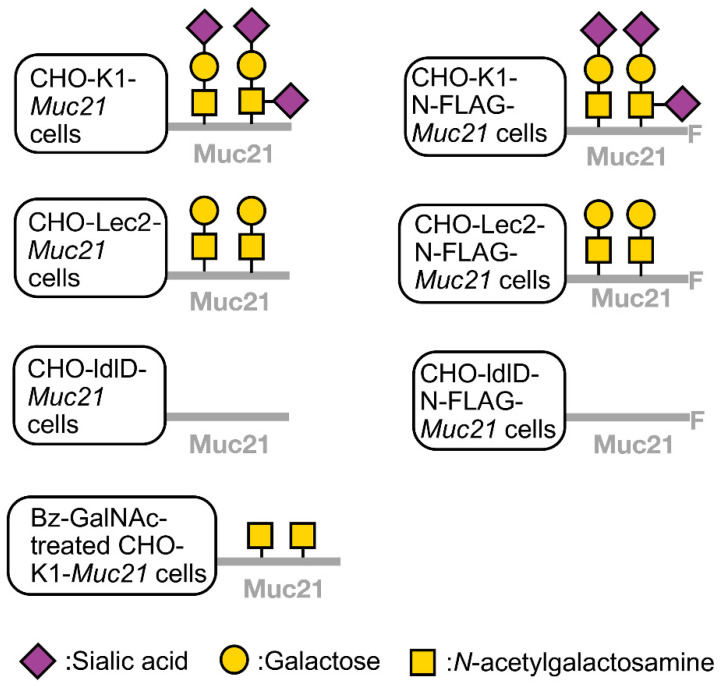
Schematics of putative glycoforms expressed by variants of Chinese hamster ovary (CHO) cells transfected with N-FLAG-*Muc21.* Muc21 with sialylated Thomsen–Friedenreich (T) antigen, i.e., consisting of *N*-Acetylgalactosamine (GalNAc), galactose (Gal), and sialic acid, is expected to be expressed on CHO-K1-pcDNA3.1-N-FLAG-*Muc21* cells. Muc21 with T-antigen, i.e., GalNAc and Gal, is expected to be produced by CHO-Lec2-pCAGGS-N-FLAG-*Muc21* cells because Lec2 cells cannot elongate sialic acid on their carbohydrate chain due to the downregulation of CMP-sialic acid Golgi transporter [12]. Non-*O*-glycosylated Muc21 is expected to be produced by CHO-ldlD-pCAGGS-N-FLAG-*Muc21* cells because ldlD cells lack the ability to produce Gal and GalNAc due to the downregulation of UDP-Gal/UDP-GalNAc 4 epimerase [12].

**Figure 3 ijms-23-06718-f003:**
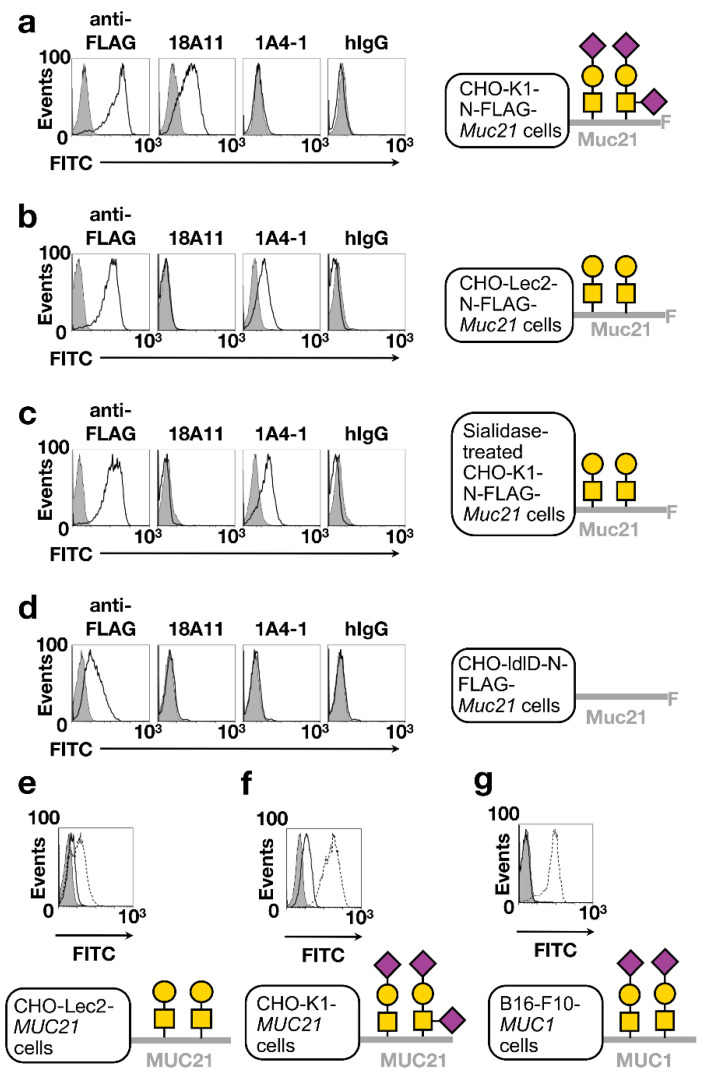
Binding of mAbs 1A4-1 and 18A11 to Muc21 transfectants expressing different glycoforms. (**a**–**d**) Flow cytometric analysis with anti-FLAG mAb, mAb 18A11, mAb 1A4-1 and hamster IgG. In each case, mock transfectants were used as controls. Control cells received the same enzyme treatment as *Muc21* transfectants. Shaded area represents antibody binding to control. Black line represents antibody binding to *Muc21* transfectants. (**a**) CHO-K1-pcDNA3.1-N-FLAG-*Muc21* cells. (**b**) CHO-Lec2-pCAGGS-N-FLAG-*Muc21* cells. (**c**) Sialidase-treated CHO-K1-pcDNA3.1-N-FLAG-*Muc21* cells. (**d**) CHO-ldlD-pCAGGS-N-FLAG-*Muc21* cells. (**e**) CHO-Lec2-pCAGGS-*MUC21* cells. Shaded area represents hamster IgG, black line mAb 1A4-1, and dashed black line mAb heM21C. (**f**) CHO-K1-pcDNA3.1-*MUC21* cells. Shaded area represents hamster IgG, black line mAb 18A11, and dashed black line mAb heM21C (specific to Tn, T, and sialyl T-MUC21). (**g**) B16-F10-*MUC1* cells. Shaded area represents hamster IgG, black line mAb 18A11, and dashed black line mAb MY.1E12 (specific to sialyl T-MUC1).

**Figure 4 ijms-23-06718-f004:**
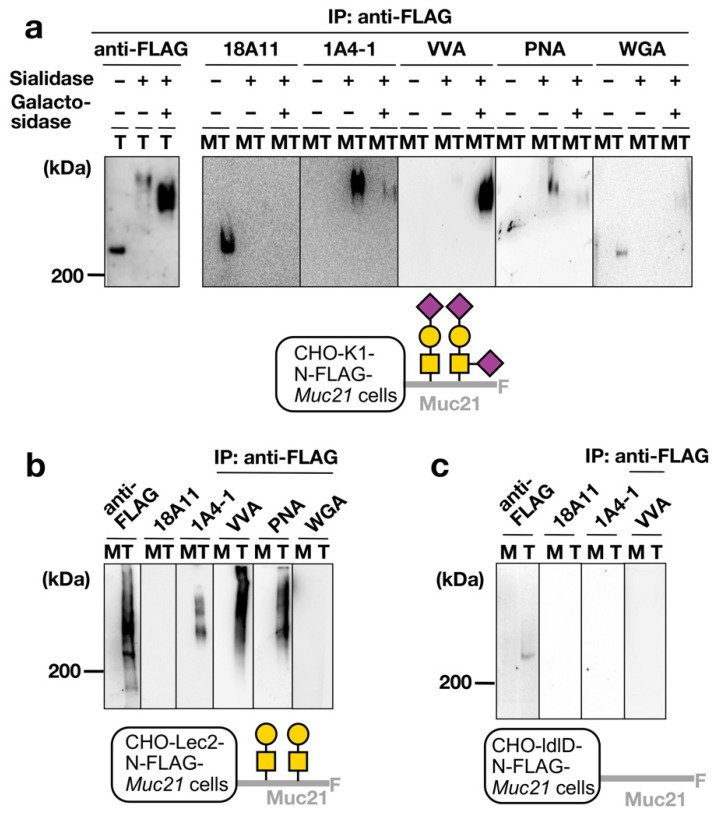
Western blotting and lectin blotting analysis results. Binding of mAb 1A4-1 and mAb 18A11 to Muc21 transfectants expressing different glycoforms. (**a**–**c**) Western blotting (WB) and lectin blotting (LB) analysis with anti-FLAG mAb, mAb 18A11, mAb 1A4-1, *Vicia villosa* lectin (VVA), *Arachis hypogaea* lectin (PNA) and *Wheat germ agglutinin* (WGA). As indicated by + (treatment) and − (no treatment), cells were treated with sialidase or both sialidase and β-galactosidase. All lysates were immunoprecipitated (IP) with anti-FLAG-M2 affinity gel before SDS-PAGE. (**a**) WB and LB analysis with the lysate of CHO-K1-pcDNA3.1-N-FLAG-*Muc21* cells (T) and the lysate of CHO-K1-pcDNA3.1 mock cells (M). (**b**) WB and LB analysis with the lysate of CHO-Lec2-pCAGGS-N-FLAG-*Muc21* cells (T) and the lysate of CHO-Lec2-pCAGGS mock cells (M). (**c**) WB and LB analysis with the lysate of CHO-ldlD-pCAGGS-N-FLAG-*Muc21* cells (T) and the lysate of CHO-ldlD-pCAGGS mock cells (M).

**Figure 5 ijms-23-06718-f005:**
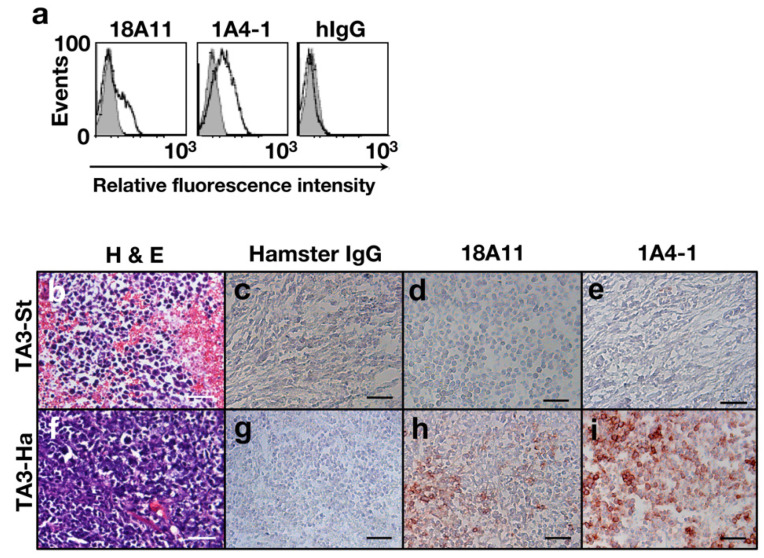
Characterization of TA3 mammary carcinoma cells and breast cancer tissue generated via the injection of TA3 cells into mammary fat pads of 5-week-old A/J mice. (**a**) FACS analysis with (**left**) mAb 18A11, (**middle**) mAb 1A4-1, and (**right**) hamster IgG. In each case, the shaded area represents antibody binding to TA3-St cells and the black line represents antibody binding to TA3-Ha cells. (**b**) Hematoxylin and eosin (H&E) staining of breast cancer tissue generated through the injection of TA3-St cells. (**c**–**e**) Tissue sections of breast cancer generated from TA3-St cells stained with (**c**) hamster IgG, (**d**) mAb 18A11, and (**e**) mAb 1A4-1. (**f**) H&E staining of breast cancer tissue generated via the injection of TA3-Ha cells. (**g**–**i**) Tissue sections of breast cancer generated from TA3-Ha cells stained with (**g**) hamster IgG, (**h**) mAb 18A11, and (**i**) mAb 1A4-1. (**b**–**i**) Scale bar: 50 μm.

**Figure 6 ijms-23-06718-f006:**
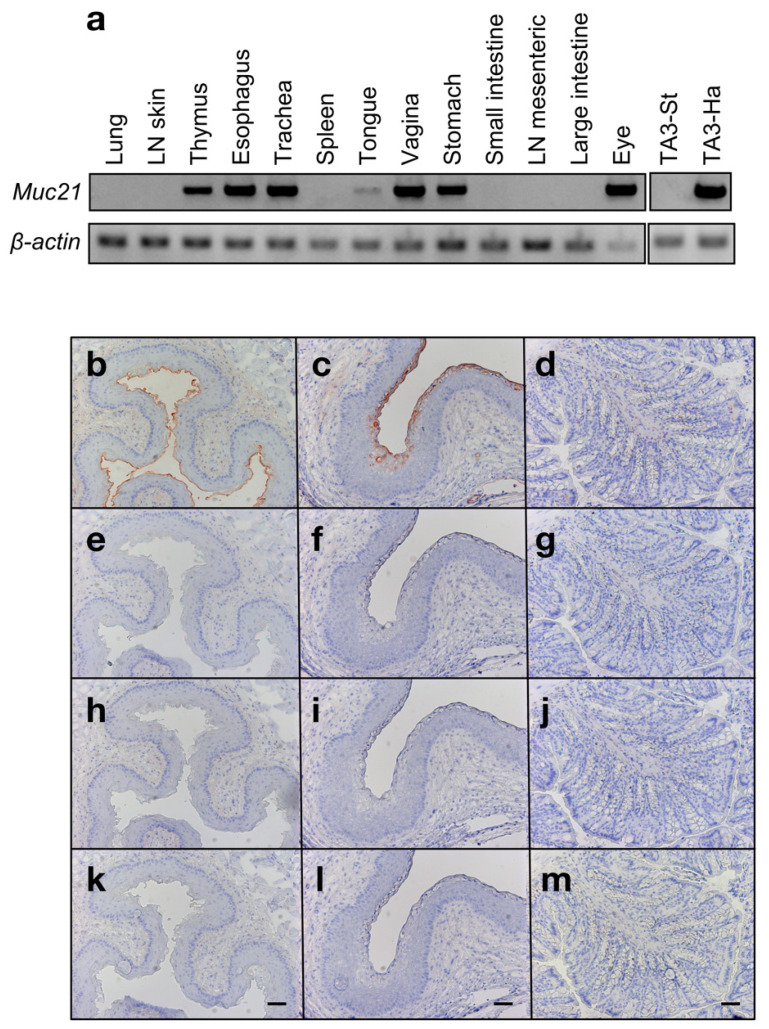
*Muc21* mRNA expression in a panel of normal mouse tissues and immunohistochemical staining of selected normal mouse tissues with mAb 1A4-1 (**a**) *Muc21* mRNA expression in a panel of C57BL/6J mouse tissues. TA3-St cells and TA3-Ha cells were used as controls (**b**–**d**) Sialidase-treated mouse tissue sections stained with mAb 1A4-1. (**b**) Esophagus. (**c**) Vagina. (**d**) Large intestine. (**e**–**g**) Untreated mouse tissue sections stained with mAb 1A4-1. (**e**) Esophagus. (**f**) Vagina. (**g**) Large intestine. (**h**–**j**) Sialidase-treated mouse tissue sections stained with hamster IgG. (**h**) Esophagus. (**i**) Vagina. (**j**) Large intestine. (**k**–**m**) Untreated mouse tissue sections. Buffer (no primary antibody) control. (**k**) Esophagus. (**l**) Vagina. (**m**) Large intestine. (**b**–**m**) Scale bar: 50 μm. LN; lymph node.

## Data Availability

All data generated or analyzed during this study are included in this published article.

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
