# Peer review of "Unique Glycoform-Dependent Monoclonal Antibodies for Mouse Mucin 21"

_ijms, 2022, doi:10.3390/ijms23126718_

Round 1

Reviewer 1 Report

In their manuscript, Nishida J., et al. report on the generation and validation of two monoclonal antibodies that recognize distinct glycoforms of mouse Muc21. The study is of great relevance and interest to the field and is scientifically sound. The present paper is extremely well written. The experiments, the methods, results and rationales are clearly presented and carefully explained. All conclusions presented by the authors are well supported by the data shown. I have no recommendations to improve this meticulous manuscript and endorse publication in its present form.

Reviewer 2 Report

This is an interesting manuscript describing two new monoclonal antibodies which recognize Muc21; one requires sialic acid, the other does not. Albeit similar MoAbs do exist, these are 

To increase clarity, I recommend that the sugar symbols should be color-coded, according to Varki A, Cummings RD, Esko JD, Freeze HH, Stanley P, Bertozzi CR, Hart GW and Etzler ME (eds) (2009). Essentials of Glycobiology. 2nd edn. Cold Spring Harbor (NY): Cold Spring Harbor Laboratory Press.

Minor comments:

page 4, line 133: space is missing after "flow".

page 4, line 148: I would change the sentence to: "These cells express glycans without sialic acids, so T antigens are present". Please explain the T antigen is Thomsen-Friedenreich antigen.

page 6, line 187: galactosidase from B. ciculans is a beta-galactosidase, please indicate it in the text (also in M&M).

page 7/213: Please change "The bindings" to "The ability of mAbs 1A4-1 and 18A11 to bind Muc21...".

page 9, line 263: please add "antigens" after T, Tn.

page 9, line 264: I recommend to add "antigens" after t, tn (i.e. "to T and Tn antigens, but not to non-O-glycolsytaed Muc21).

page 10, line 282: please change "will" to "amy possibly".

page 10, line 303: please delete "highly".

page 10, line 305: Please add "Presence of" at the biginning of the sentence.
